# Use of the Visceral Adiposity Index as an Indicator of Chronic Kidney Disease in Older Adults: Comparison with Body Mass Index

**DOI:** 10.3390/jcm11216297

**Published:** 2022-10-26

**Authors:** Bokun Kim, Gwon-Min Kim, Sechang Oh

**Affiliations:** 1Department of Anti-Ageing Health Care, Changwon National University, Changwon 51140, Korea; 2Future Convergence Research Institute, Changwon National University, Changwon 51140, Korea; 3Department of Sports Healthcare, In-Je University, Gimhae 50834, Korea; 4Medical Research Institute, Pusan National University, Busan 46241, Korea; 5Faculty of Rehabilitation, R Professional University of Rehabilitation, Tsuchiura 300-0032, Japan

**Keywords:** chronic kidney disease, body mass index, older adults, visceral adiposity index

## Abstract

The visceral adiposity index (VAI) was recently introduced to quantify visceral fat accumulation and dysfunction. This cross-sectional study explored whether the VAI is associated with chronic kidney disease (CKD) in older adults and compared its utility with that of body mass index (BMI) for predicting CKD. In total, 7736 older adults (3479 men and 4257 women) aged ≥ 60 years were divided into normal, mild, and moderate-to-severe CKD groups. Associations of the VAI and BMI with CKD were compared among the groups, and cut-off points for moderate-to-severe CKD (^MS^CKD) were established. While the VAI could discriminate among all of the groups, the BMI could not. The severity of CKD was more strongly associated with the VAI than BMI. The odds ratios indicated that, in the fully adjusted model, the VAI was a significant predictor of ^MS^CKD in both men and women, while the BMI was a significant predictor only in men. For the VAI, the area under the receiver operating characteristic curve values for men and women were 0.631 (cut-off point: ≥2.993) and 0.588 (≥4.001), compared with 0.555 (≥25.335) and 0.533 (≥24.096) for BMI, respectively. Taken together, the findings suggest that the VAI is associated with CKD and represents a better indicator for the disease than BMI.

## 1. Introduction

Chronic kidney disease (CKD) is closely associated with aging and increases the likelihood of mortality, hypertension, and type 2 diabetes [1,2,3]. In the US, CKD prevalence rates of 5.7%, 8.9%, and 33.2% were reported for people aged 20–39, 40–59, and ≥60 years, respectively [4]. Correspondingly, CKD prevalence rates in South Korea of 8.2%, 13.7%, and >30.0% were reported for people aged 20–34, 35–59, and ≥60 years, respectively [5,6]. In 2017, CKD reportedly caused 1.2 million deaths worldwide, which is expected to increase to 2.2 million by 2040 in the best-case scenario, and to 4.0 million in the worst-case scenario [7]. The CKD prevalence and death rates are predicted to increase because of the aging of the worldwide population. Given that CKD prevalence rates are highest among elderly populations and that CKD increases the mortality rate in this group, significant improvements in the screening, detection, and treatment of the disease are needed to prevent or slow the progression to severe CKD, especially in high-risk individuals.

Obesity is another significant public health concern and its relationship with CKD has been well-documented [8,9,10,11]. Previous studies analyzed the CKD risk according to body mass index (BMI) [12,13,14]. However, despite its wide use as an indicator of obesity and body adiposity, BMI does not clearly reflect overall fat mass; in particular, it cannot discriminate visceral fat, which is responsible for many health problems [15,16,17,18]. In recent studies, increased fat mass percentage was strongly associated with CKD [8,19,20]. Therefore, better measures of fat mass than the BMI are needed to facilitate CKD management [19]. Because expensive equipment or extensive measurements are currently required to assess fat mass, a simple, inexpensive, and generally applicable method should be established to identify individuals at high risk of CKD.

A new body adiposity index, the visceral adiposity index (VAI), was recently introduced. The VAI is based on the BMI, waist circumference, and triglyceride (TG) and high-density lipoprotein cholesterol (HDLC) levels, and is used as an indicator of visceral fat accumulation and dysfunction [21]. The validity and reliability of the VAI have been demonstrated in several ethnic groups and countries. The VAI was a highly accurate predictor of type 2 diabetes in the Qatari population [22]. As well as reflecting an unhealthy metabolic phenotype, the VAI was a surrogate indicator of arterial stiffness in the Chinese population [23]. Moreover, the VAI was significantly related to fibrosis in an Italian population with non-alcoholic fatty liver disease [24]. However, despite the proven utility of the VAI as an indicator of several health conditions, few cross-sectional investigations have examined the relationship between the VAI and CKD [25]. Furthermore, even though the prevalence of CKD in the older population is more than twice as high as that in middle-aged individuals, a limited subset of the latter population has been analyzed [26,27,28].

This cross-sectional study was conducted to investigate the relationship between the VAI and CKD in the older Korean population. Additionally, the utility of the VAI for predicting CKD was compared with that of the BMI.

## 2. Materials and Methods

### 2.1. Study Design and Subjects

General health, nutrition and lifestyle data from the Korea National Health and Nutritional Examination Survey (KNHANES; 2015–2019) were analyzed. Of the 11,192 older adults (aged ≥ 60 years) in the 2015–2019 KNHANES, 7736 (3479 males and 4257 females) were included in our analysis. A flow diagram of subject recruitment is presented in Figure 1. All subjects provided written informed consent, and the study was conducted in accordance with the principles of the Declaration of Helsinki and approved by the Institutional Review Board of Silla University (1041449-202203-HR-001).

### 2.2. Body Adiposity Indices and Estimated Glomerular Filtration Rate (eGFR)

Waist circumstance (WC) and height were measured to the nearest 0.1 cm; body mass was measured to within 0.1 kg with a digital electronic scale (JENIX DS-102; Dong Sahn Jenix Co., Seoul, Korea). Blood samples were collected in the morning after an 8 h fast. The TG, HDLC, and creatinine levels were measured using enzymatic and homogeneous enzymatic colorimetric methods, along with the Jaffe rate-blanked creatinine assay, and validated at a certified laboratory (Seegene Medical Foundation, Seoul, Korea).

The BMI was calculated as body mass (kg)/height (m^2^), and the VAI as (WC/(36.58 + (1.89 × BMI)) × ((TG/0.81) × (1.52/HDLC)) for women and (WC/(39.68 + (1.88 × BMI)) × ((TG/1.03) × (1.31/HDLC)) for men [21]. The BMI and VAI were converted into Z-scores using the following equation: (individual value − mean value)/standard deviation (SD) [29,30]. The ability of the two body adiposity indices to predict CKD was then analyzed.

The eGFR was defined in accordance with the Japanese version of the Modification of Diet in Renal Disease study equation: eGFR (mL/min/1.73 m^2^) = 194 × serum creatinine^−1.094^ × age^−0.287^ × 0.739 (for women) [19,28,30,31]. The subjects were divided into three groups on the basis of eGFR tertile: normal (N) group, eGFR ≥ 60.0 mL/min/1.73 m^2^; mild CKD (^M^CKD) group, eGFR = 45.0–59.9 mL/min/1.73 m^2^; and moderate-to-severe CKD (^MS^CKD) group, eGFR < 45.0 mL/min/1.73 m^2^ [19,29,31,32].

### 2.3. Statistical Analysis

The data are provided as mean ± SD or 95% confidence intervals (CIs), and odds ratios (ORs) and 95% CIs were calculated. The independent *t*-test and Mann–Whitney U test were used to compare the males and females. One-way analysis of variance was utilized to compare anthropometric data among the three groups, with the Bonferroni post-hoc test applied. Data that were not normally distributed were compared between groups using the Mann–Whitney U test. The Jonckheere–Terpstra test (two-tailed) was utilized to generate standardized statistics allowing comparison of trends among the three groups [19,29,33,34]. *p*-values < 0.05 were considered significant in these analyses. Logistic regression was used to evaluate the associations of the VAI and BMI Z-scores with ^MS^CKD. In the fully adjusted model, potential confounders such as education level, household income, smoking, drinking, handgrip strength, moderate-to-vigorous physical activity, and nutritional factors known or suspected to influence associations with ^MS^CKD were controlled for. SPSS software (ver. 20.0; IBM Corp., Armonk, NY, USA) was employed for the statistical analyses. The optimal cut-off points for the VAI and BMI in male and female subjects to predict CKD were derived from receiver operating characteristic (ROC) curve analysis, i.e., area under the ROC curve (AUC) values. Sensitivity and specificity were also calculated. The analysis was carried out using MedCalc for Windows software (ver. 9.1.0.1; MedCalc, Ostend, Belgium).

## 3. Results

Table 1 lists the characteristics of the study subjects and differences between males and females. The mean eGFR values for the entire cohort and for the male and female subjects were 63.3 ± 19.3, 73.5 ± 19.8, and 54.9 ± 14.1 mg/dL, respectively; the mean VAI scores were 4.89 ± 3.47, 3.87 ± 2.84, and 5.73 ± 3.71, respectively; finally, the mean BMI values were 24.23 ± 3.14, 24.20 ± 3.22, and 24.25 ± 3.07 kg/m^2^, respectively. Significant differences between male and female subjects were observed in the association between the eGFR and VAI (*p* < 0.001), but not for the association between the eGFR and BMI. Additional data for the entire cohort and for the male and female subjects are provided in the Appendix A.

Sex-specific differences and trends in the VAI and BMI according to eGFR category are presented in Figure 2. For the male subjects, the trend test revealed a significant decreasing tendency in the eGFR across the groups (from N to ^MS^CKD; standardized statistic (SS) = −46.75, *p* < 0.001), while the opposite tendency was observed for the VAI and BMI (SS = 8.25 and 2.78, respectively; *p* < 0.01 for both). Post-hoc tests demonstrated that the VAI could discriminate among the N, ^M^CKD, and ^MS^CKD groups. The BMI could distinguish between the N and ^MS^CKD groups, but not between the N and ^M^CKD or ^M^CKD and ^MS^CKD groups. For the female subjects, the trend test revealed a significant decreasing tendency in the eGFR across groups (from N to ^MS^CKD; SS = −68.56; *p* < 0.001), while the opposite tendency was found for the VAI and BMI (SS = 9.78 and 2.48, respectively; *p* < 0.05 for both). Post-hoc tests revealed that the VAI could discriminate among the N, ^M^CKD, and ^MS^CKD groups. The BMI could distinguish between the N and ^M^CKD groups and N and ^MS^CKD groups, but not between the ^M^CKD and ^MS^CKD groups. Table 2 shows the sex differences in parameters essential to calculate the VAI, BMI, and eGFR, such as the waist circumference, height, weight, BMI, TG, HDLC, creatinine levels, and age. Appendix A shows the results of analyses of additional parameters.

The associations of the VAI and BMI Z-scores with ^MS^CKD are shown in Figure 3. Males and females were divided into tertiles based on the VAI Z-scores. In the unadjusted model, compared with the lowest tertile, the highest and middle male and female tertiles had ORs of 3.39 (95% CI: 2.27–5.06) and 2.05 (95% CI: 1.34–3.14), and 2.43 (95% CI: 2.00–2.95) and 1.71 (95% CI: 1.40–2.08), respectively, for ^MS^CKD. In the fully adjusted model, compared with the lowest tertile, the highest and middle male and female tertiles had ORs of 3.19 (95% CI: 2.12–4.79) and 2.10 (95% CI: 1.37–3.24), and 2.41 (95% CI: 1.94–3.00) and 1.77 (95% CI: 1.42–2.21), respectively. Regarding the BMI Z-scores, in the unadjusted model, compared with the lowest tertile, the highest and middle male and female tertiles had ORs of 1.62 (95% CI: 1.13–2.31) and 1.16 (95% CI: 0.79–1.70), and 1.26 (95% CI: 1.04–1.52) and 1.16 (95% CI: 0.95–1.40), respectively, for ^MS^CKD. In the fully adjusted model, compared with the lowest tertile, the highest and middle male and female tertiles had ORs of 1.47 (95% CI: 1.02–2.11) and 1.09 (95% CI: 0.74–1.61), and 1.22 (95% CI: 0.98–1.51) and 1.01 (95% CI: 0.81–1.26), respectively, for ^MS^CKD.

Figure 4 shows the ROC curves of the VAI and BMI plotted against ^MS^CKD for each sex. For the VAI, the optimal cut-off points (in terms of the balance between sensitivity and specificity) were ≥2.993 and ≥4.001 for males and females, respectively (*p* < 0.001 for both). At these cut-off points, the sensitivity and specificity were 49.15% and 72.82% for males, and 43.91% and 71.04% for females, respectively. For the BMI, the optimal cut-off points were ≥25.335 and ≥24.093 in males and females, respectively (*p* < 0.01 for both). At these cut-off points, the sensitivity and specificity were 68.18% and 42.05% for males, and 52.20% and 54.46% for females, respectively.

## 4. Discussion

This study suggests that the VAI is a better indicator of CKD than BMI. First, the VAI could discriminate among all groups (i.e., the N, ^M^CKD, and ^MS^CKD groups), whereas the BMI could not. Second, in the trend analysis, the SS values for the VAI were higher than those for BMI. Third, in the fully adjusted model, the VAI was significantly associated with ^MS^CKD in both male and female subjects, while for BMI, this was only the case for males. Lastly, the AUC values for males and females were higher for the VAI than BMI. These findings show that the VAI is related to CKD more strongly than BMI. Thus, we believe that the VAI, which can be used to evaluate visceral fat accumulation and dysfunction, is a superior indicator of CKD than BMI.

As people age, adverse changes in body composition tend to occur, such as gradual muscle mass reduction and increased fat mass (especially visceral fat) [35,36,37]. Such changes are more closely linked to negative health outcomes than BMI, which does not reflect overall fat mass or discriminate visceral fat; the latter has attracted attention as a primary cause of poor health [16,19,38]. For decades, the fat cell was considered a storehouse of energy, but in recent years, its function as an endocrine organ has been highlighted [39,40,41]. As excessive fat accumulates, proinflammatory cytokines, including tumor necrosis factor-alpha (TNF-α) and interleukin-6 (IL-6), are produced in excess, while the expression of anti-inflammatory adipokines such as adiponectin is reduced [26]. Additionally, increased visceral fat cell volume promotes IL-6 secretion via increased free fatty acids [26]. This signaling cascade increases the production of macrophage chemotactic factors, which leads to macrophage infiltration and excessive TNF-α production [26]. Finally, increased macrophage infiltration and TNF-α production lead to chronic inflammation and a decline in kidney function [42,43,44]. Thus, given the limitations of BMI, it is paramount to evaluate the performance of the VAI as an indicator of CKD risk.

Regarding the VAI, the odds of CKD of the middle and highest male and female tertiles were significantly higher than those of the subjects in the lowest tertile, in both the unadjusted and fully adjusted models constructed in this study. For the BMI, this was only the case in the fully adjusted model; in the unadjusted model, significance was seen for males but not females. In addition, the OR of CKD in the analysis of the VAI was higher than that in the analysis of BMI for male subjects. These findings demonstrated that the predictive capacity of the VAI for CKD is better than that of BMI, which may not be appropriate to evaluate the risk of CKD despite its wide use for this purpose.

The results of the AUC analysis shown in Figure 4, i.e., the sensitivity and specificity values, support the notion that the VAI is superior to BMI for predicting CKD. Xu et al. (2016) and Xiao et al. (2020) suggested VAI cut-off values for CKD of 1.47, 2.208, and 2.177 for both sexes, males, and females, respectively [26,28]. However, their cut-off points were based on quartiles rather than tertiles. Additionally, Xu et al. (2016) did not devise separate cut-offs for males and females, and the number of subjects was relatively small in both studies [28]. Therefore, we consider the cut-off points in this study to be more appropriate.

There were several limitations to the current study. Although factors potentially influencing the relationship between the VAI and CKD, such as demographic and lifestyle factors, were adjusted for, the cross-sectional design did not allow firm conclusions to be drawn regarding the predictive capacity of the VAI for CKD. Enough longitudinal examination is needed to validate the findings of the current study [45]. Moreover, although our findings coincide with previous studies, they may generalize to other ethnicities or regions. Thus, studies of non-Asian populations should be carried out to further explore the relationship between the VAI and CKD. Finally, as we included only older adults in this study, studies of young and middle-aged populations are needed.

## 5. Conclusions

The VAI discriminated among all of the groups (N, ^M^CKD, and ^MS^CKD) in this study, whereas the BMI did not. Additionally, the strength of the association between the VAI and CKD was greater than that seen for BMI in the trend test. Furthermore, the odds of ^MS^CKD were significantly higher in the middle and highest VAI tertiles compared with the lowest one for both males and females; however, for the BMI, this was only the case in male subjects in the fully adjusted analysis. Finally, the AUC values for males and females were higher for the VAI than BMI. Taken together, the results show that the VAI is a better indicator of CKD than BMI.

## Figures and Tables

**Figure 1 jcm-11-06297-f001:**
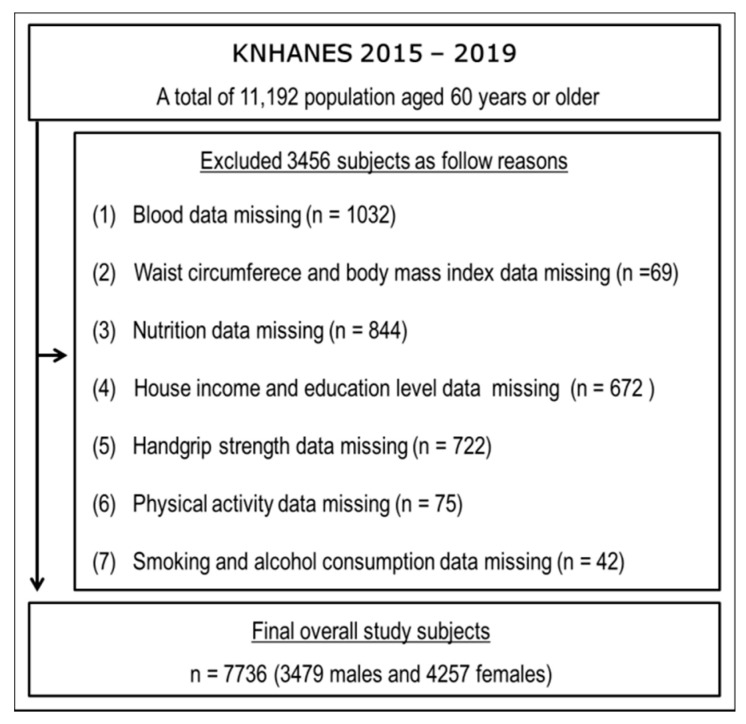
Flow chart of study subjects.

**Figure 2 jcm-11-06297-f002:**
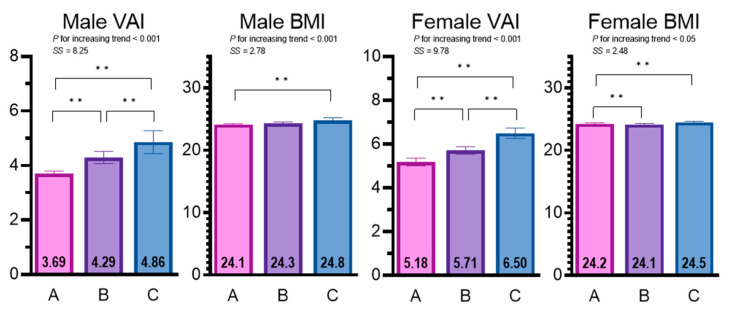
Sex-specific differences and trends in the VAI and BMI according to eGFR category. Values are mean ± 95 confidence interval. A = eGFR ≥ 60 (95% CI); B = eGFR 45–59.9 (95% CI); C = eGFR < 45; VAI = visceral adiposity index; BMI = body mass index; SS = standardized statistic. ** *p* < 0.01 for the difference between groups.

**Figure 3 jcm-11-06297-f003:**
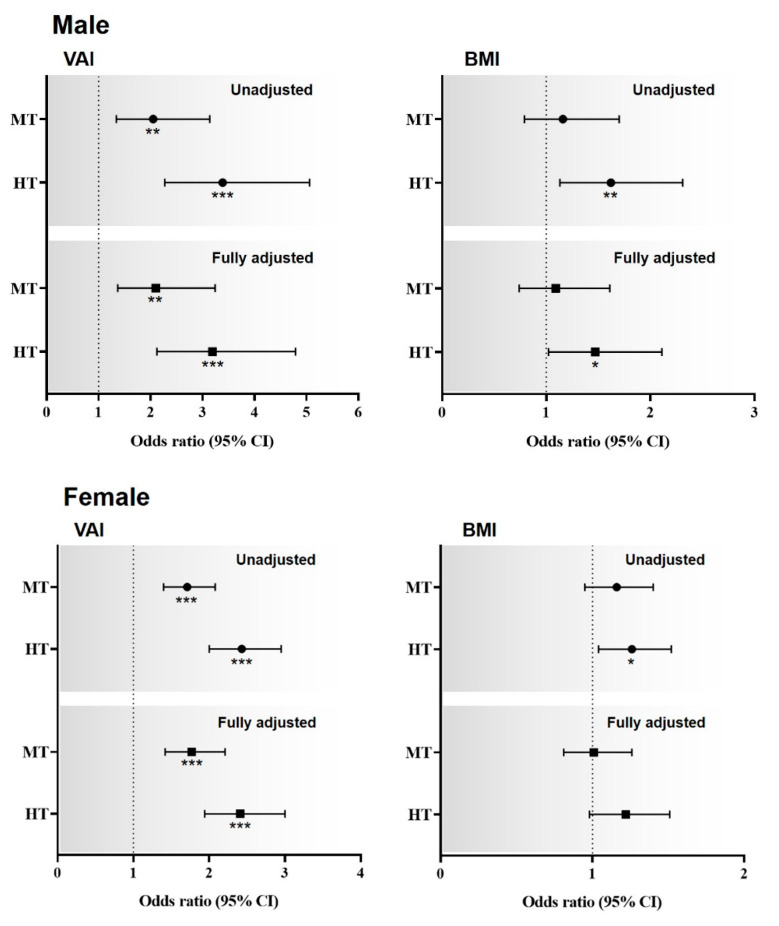
Sex-specific odds ratios for the relationships of ^MS^CKD with the VAI and BMI Z-scores. Dotted line: reference; solid line: 95% confidence interval; Black circle: odds ratio. * *p* < 0.05, ** *p* < 0.01, *** *p* < 0.001 for the odds ratios for ^MS^CKD, compared with the lowest tertiles. Abbreviations: HT, highest tertile; MT, middle tertile. VAI = visceral adiposity index; BMI = body mass index.

**Figure 4 jcm-11-06297-f004:**
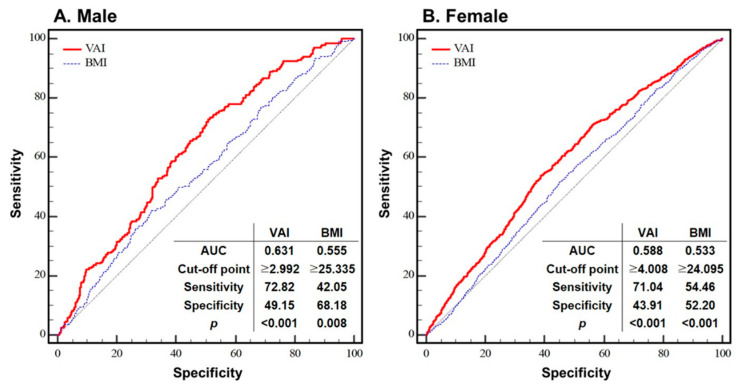
The ROC curves of the VAI and BMI plotted against ^MS^CKD for each sex. Thin black line: reference; solid red line: area under the curve (AUC), indicating the accuracy of the VAI for the identification of ^MS^CKD; thin dotted blue line: area under the curve (AUC), indicating the accuracy of the BMI for the identification of ^MS^CKD; cut-off point: the value of VAI or BMI that predicts ^MS^CKD; sensitivity: the probability of individuals who actually have ^MS^CKD to be predicted to have ^MS^CKD; specificity: the probability of individuals who do not have ^MS^CKD to be predicted not to have ^MS^CKD. Abbreviation: VAI = visceral adiposity index; BMI = body mass index; AUC = area under curve.

**Table 1 jcm-11-06297-t001:** The characteristics of study subjects and differences between males and females.

	Overall (*n* = 7736)	Male (*n* = 3479)	Female (*n* = 4257)	*p* Value
eGFR, mg/dL ^†^	63.3 ± 19.3	73.5 ± 19.8	54.9 ± 14.1	<0.001
Visceral adiposity index ^†^	4.89 ± 3.47	3.87 ± 2.84	5.73 ± 3.71	<0.001
Body mass index, kg/m^2 †^	24.23 ± 3.14	24.20 ± 3.22	24.25 ± 3.07	=0.266
Age, year ^†^	69.7 ± 6.4	69.7 ± 6.3	69.8 ± 6.5	=0.741
Height, cm	158.9 ± 8.8	159.0 ± 8.8	158.9 ± 8.7	=0.478
Body weight, kg ^†^	61.3 ± 10.0	61.3 ± 10.3	61.3 ± 9.8	=0.882
Waist circumstance, cm ^†^	85.6 ± 8.9	85.6 ± 9.1	85.6 ± 8.8	=0.625
Creatinine, mg/dL ^†^	0.85 ± 0.31	0.86 ± 0.33	0.84 ± 0.29	=0.053
Triglyceride, mg/dL ^†^	131.2 ± 69.1	132.4 ± 74.3	130.2 ± 64.5	=0.543
HDLC, mg/dL	49.1 ± 11.9	48.9 ± 11.9	49.2 ± 11.9	=0.215

Values are means ± SD. eGFR = estimated glomerular filtration rate; HDLC = high density lipoprotein cholesterol. ^†^ Mann–Whitney U test was applied to assess the difference between groups.

**Table 2 jcm-11-06297-t002:** Sex-specific differences, and trends of subjects by eGFR category.

	eGFR Category (mL/min/1.73 m^2^)			*p* for Difference	SS ^‡^	*p* for Trend ^‡^
	A	B	C			
Male, *n*	2594	690	195			
eGFR, mg/dL ^†^	81.5 ± 15.8 (80.8, 82.1)	54.1 ± 4.2 (53.8, 54.4)	36.6 ± 8.4 (35.4, 37.8)	A > B > C	−46.75	<0.001
Z score of VAI ^†^	−0.06 ± 0.97 (−0.10, −0.03)	0.15 ± 1.05 (0.07, 0.23)	0.35 ± 1.05 (0.20, 0.50)	A < B < C	8.25	<0.001
Z score of BMI	−0.02 ± 1.01 (−0.06, 0.02)	0.03 ± 0.97 (−0.04, 0.10)	0.18 ± 0.98 (0.04, 0.31)	A < C	2.78	<0.01
Age, year	69.4 ± 6.3 (69.2, 69.7)	70.5 ± 6.2 (70.0, 71.0)	70.6 ± 6.6 (69.7, 71.5)	A < B < C	4.47	<0.001
Height, cm ^†^	157.2 ± 8.3 (156.9, 157.6)	164.5 ± 8.1 (163.9, 165.1)	162.9 ± 8.2 (161.7, 164.0)	A < B, A < C, B > C	20.19	<0.001
Body mass, kg	59.8 ± 9.9 (59.4, 60.1)	65.9 ± 10.3 (65.1, 66.6)	65.8 ± 10.3 (64.4, 67.3)	A < B, C	15.37	<0.001
WC, cm	84.7 ± 9.1 (84.3, 85.0)	87.7 ± 8.6 (87.1, 88.3)	89.8 ± 8.8 (88.6, 91.1)	A < B < C	10.60	<0.001
Cre, mg/dL ^†^	0.75 ± 0.12 (0.74, 0.75)	1.06 ± 0.08 (1.05, 1.07)	1.67 ± 0.85 (1.55, 1.79)	A < B < C	46.52	<0.001
TG, mg/dL	130.5 ± 75.1 (127.6, 133.4)	137.0 ± 73.9 (131.5, 142.5)	142.0 ± 62.8 (133.1, 150.9)	NS	4.15	<0.001
HDLC, mg/dL	50.0 ± 12.0 (49.6, 50.5)	46.3 ± 11.3 (45.5, 47.2)	43.1 ± 10.5 (41.6, 44.6)	A < B < C	−10.08	<0.001
Female, *n*	1482	1677	1098			
eGFR, mg/dL ^†^	70.2 ± 8.6 (69.8, 70.7)	52.3 ± 4.3 (52.1, 52.5)	38.2 ± 5.8 (37.8, 38.5)	A > B > C	−68.56	<0.001
Z score of VAI ^†^	−0.15 ± 0.93 (−0.20, −0.10)	−0.01 ± 0.99 (−0.05, 0.04)	0.21 ± 1.07 (0.14, 0.27)	A < B < C	9.78	<0.001
Z score of BMI ^†^	−0.01 ± 1.04 (−0.06, 0.04)	−0.04 ± 1.00 (−0.09, 0.01)	0.07 ± 0.94 (0.02, 0.13)	A, B < C	2.48	<0.05
Age, year	69.3 ± 6.4 (69.0, 69.6)	70.0 ± 6.5 (69.6, 70.3)	70.1 ± 6.4 (69.8, 70.5)	A < B, C	3.50	<0.001
Height, cm ^†^	154.1 ± 6.5 (153.8, 154.4)	159.7 ± 8.7 (159.2, 160.1)	164.1 ± 7.8 (163.6, 164.5)	A < B < C	29.93	<0.001
Body mass, kg ^†^	57.6 ± 8.7 (57.1, 58.0)	61.6 ± 9.4 (61.1, 62.0)	66.0 ± 9.6 (65.4, 66.5)	A < B < C	21.91	<0.001
WC, cm	83.9 ± 8.7 (83.4, 84.3)	85.4 ± 8.6 (85.0, 85.8)	88.3 ± 8.5 (87.8, 88.8)	A < B < C	12.81	<0.001
Cre, mg/dL ^†^	0.64 ± 0.06 (0.64, 0.64)	0.83 ± 0.07 (0.83, 0.83)	1.14 ± 0.41 (1.12, 1.17)	A < B < C	68.08	<0.001
TG, mg/dL	126.4 ± 64.3 (123.1, 129.6)	130.2 ± 64.5 (127.1, 133.3)	135.3 ± 64.5 (131.5, 139.1)	A < C	4.21	<0.001
HDLC, mg/dL	51.6 ± 11.9 (51.0, 52.2)	49.2 ± 11.8 (48.6, 49.8)	46.0 ± 11.3 (45.3, 46.6)	A > B > C	−12.55	<0.001

Values are means ± SD (95% CI). ^†^ Mann–Whitney U test was applied to assess the difference between groups. ^‡^ Jonckheere–Terpstra test was used to assess the trend among three groups. A = eGFR ≥ 60 (95% CI); B = eGFR 45–59.9 (95% CI); C = eGFR < 45 (95% CI); Cre = Creatinine eGFR = estimated glomerular filtration rate; TG = Triglyceride; VAI = visceral adiposity index; BMI = body mass index; HDLC = high density lipoprotein cholesterol; SS = standardized statistic; NS = not significant; WC = Waist circumference.

## Data Availability

The data sets analyzed during the current study are available from the corresponding author on reasonable request.

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
