# Peer review of "Use of the Visceral Adiposity Index as an Indicator of Chronic Kidney Disease in Older Adults: Comparison with Body Mass Index"

_jcm, 2022, doi:10.3390/jcm11216297_

Round 1

Reviewer 1 Report

Thank you for your paper quite difficult to read although well written just because of the amount of statistics I only have one minor comment instead of senior I would replace throughout and be consistent using older as the term for this population

Reviewer 3 Report

Dear Authors,

I read the manuscript with a great interest. In my opinion it is a reliable study confirming the usefulness of visceral adiposity index in the chronic kidney disease prediction.  However, I suggest including in the "Introduction" section some findings confirming VAI in prediction of incident CAD (for example Bamba et al. The Visceral Adiposity Index Is a Predictor of Incident Chronic Kidney Disease: A Population-Based Longitudinal Study.

Moreover, I feel that results not clearly confirming VAI as a CKD predictor should be presented in the "Discussion" section (for example Dai et al. Visceral Adiposity Index and Lipid Accumulation Product Index: Two Alternate Body Indices to Identify Chronic Kidney Disease among the Rural Population in Northeast China or Mousapour et al. Predictive performance of lipid accumulation product and visceral adiposity index for renal function decline in non-diabetic adults, an 8.6-year follow-up).
